# Benefit-risk analysis of maintaining essential Reproductive, Maternal, Newborn, and Child Health (RMNCH) services against risk of COVID-19 infection

Eva Weissman[1], Denise Buchner[2], Nilmini Hemachandra[3], Khalid Siddeeg[3], Mohammad Samim Soroush[4], Ahmed Javed Rahmanzai[4], Paata Chikvaidze[5], Zaid Muayad Yassen[6], Hanan Hasan[7], Mohamed Berraho[8], Nouzha Dghoughi[9], Hachri Hafid[10], Raza Mahmood Zaidi[11], Wahaj Zulfiqar[11], Sayema Awes[11], Atiya Aabroo[11], Qudsia Uzma[12], Mohamed Abubakar Hagi Mohamed Fiidow[13], Abdulkadir Wehliye Afrah[14], Abdullahi Abdulle Ali[13], Abdulazim Ali Awadalla[15], Muntasir Mohammed Osman EL Hassan[16], Esmehan Babeker Elkheir[15], Theresa Diaz[17] *

1 Maternal, Newborn, Child and Adolescent Health and Ageing Department, World Health Organization, Consultant, New York, New York, United States of America, 2 Reproductive and Maternal Health Unit, World Health Organization Regional Office for the Eastern Mediterranean, Consultant, Calgary, Canada, 3 Reproductive and Maternal Health Unit, WHO Regional Office for the Eastern Mediterranean, Cairo, Egypt, 4 Emerging Leaders Consulting Services, Kabul, Afghanistan, 5 World Health Organization Country Office, Kabul, Afghanistan, 6 University of Mosul College of Medicine, Mosul, Iraq, 7 World Health Organization Country Office, Baghdad, Iraq, 8 Sidi Mohammed Ben Abdellah University, Fez, Morocco, 9 Population Directorate Ministry of Health, Rabat, Morocco, 10 World Health Organization Country Office, Rabat, Morocco, 11 Ministry of National Health Services, Regulation & Coordination, Islamabad, Pakistan, 12 World Health Organization Country Office Pakistan, Islamabad, Pakistan, 13 Capital University of Somalia, Mogadishu, Somalia, 14 Federal Ministry of Health Somalia, Mogadishu, Somalia, 15 World Health Organization Country Office Sudan, Consultant, Khartoum, Sudan, 16 Epidemiology Directorate, Federal Ministry of Health Sudan, Khartoum, Sudan, 17 Maternal, Newborn, Child and Adolescent Health and Ageing Department, World Health Organization, Geneva, Switzerland

* tdiaz@who.int

**Data Availability Statement:** The six country models data can be accessed at: https://doi.org/10.6084/m9.figshare.14768901.v1.

## Abstract

With the COVID-19 pandemic spreading across the world, its disruptive effect on the provision and utilization of non- COVID related health services have become well-documented. As countries developed mitigation strategies to help continue the delivery of essential health services through the pandemic, they needed to carefully weigh the benefits and risks of pursuing these strategies. In an attempt to assist countries in their mitigation efforts, a Benefit-Risk model was designed to provide guidance on how to compare the health benefits of sustained essential reproductive, maternal, newborn and child (RMNCH) services against the risk of SARS-CoV-2 infections incurred by the countries' populations when accessing these services. This article describes how two existing models were combined to create this model, the field-testing process carried out from November 2020 through March 2021 in six countries and the findings. The overall Benefit-Risk Ratio in the 6 countries analyzed was found to be between 13.7 and 79.2, which means that for every 13.7 to 79.2 lives gained due to increased RMNCH service coverage, there was one loss of a life related to COVID-19. In all cases and for all services, the benefit of maintaining essential health services far exceeded the risks associated with additional COVID-19 infections and deaths. This

**Funding:** This work was supported, in whole or in part, by the Bill & Melinda Gates Foundation [SRMNCAH COVID Risk Benefit modelling – Covid Mitigation Grant INV-017424]. Under the grant conditions of the Foundation, a Creative Commons Attribution 4.0 Generic License has already been assigned to the Author Accepted Manuscript version that might arise from this submission. The funders had no role in study design, data collection and analysis, decision to publish, or preparation of the manuscript.

**Competing interests:** The authors have declared that no competing interests exist.

modelling process illustrated how essential health services can continue to operate during a pandemic and how mitigation measures can reduce COVID-19 infections and restore or increase coverage of essential health services. Overall, this Benefit-Risk analysis underscored the importance and value of maintaining coverage of essential health services even during public health emergencies, including the recent COVID-19 pandemic

## Introduction

With the COVID-19 pandemic affecting more and more countries, its disruptive effect on the provision and utilization of non- COVID-19 related health services have become well-documented [1,2]. On the supply side, immunization and malaria campaigns had been suspended [3–5]. Health staff originally dedicated to maternal or child health care were reassigned to provide care to COVID-19 patients or became unavailable as they themselves fall ill with the disease [6,7]. Interrupted supply chains led to shortages of essential drugs and commodities [8]. On the demand side, patients' fear of infection with SARS-CoV-2 had resulted in a decrease in the use of health services across the entire spectrum of health services [1,2].

Over the last year, a growing number of modelers began to use their existing models to assess the impact of the COVID-19 pandemic on non-COVID-19 essential health services and to demonstrate the impact these service disruptions had on mortality and morbidity at country-level [9–11]. While the projections of the large number of maternal and child lives lost due to the disruption of health services has helped direct attention to the importance of maintaining these essential health services, countries have faced challenges when trying to actually restore coverage of these services.

To offset the potential indirect impact of the COVID-19 pandemic due to health service disruptions the World Health Organization in June 2020, published interim guidelines for countries seeking to maintain essential health services during the pandemic [12]. The mitigation strategies proposed tackle supply and/or demand factors. They include approaches such as bundling the delivery of several interventions in one visit, prioritizing high-risk cases, providing several months of supplements or contraceptives at a time, or shifting tasks to lower cadres of health workers. Establishing safe and efficient patient flow and providing personal protective equipment (PPE) to all health workers can reduce the risk of infection to health workers and patients alike. The suspension of co-payments or user fees may reduce financial barriers to access.

As mitigation strategies were being formulated to help continue the delivery of essential health services through the COVID-19 pandemic, countries needed to carefully weigh the benefits and risk of pursuing these strategies. WHO and global partners in reproductive, maternal, newborn, and child health (RMNCH), from August through September of 2020, in an attempt to assist countries in their mitigation efforts developed a Benefit-Risk model designed to provide guidance on how to compare the health benefits of sustained essential RMNCH services against the risk of SARS-CoV-2 infections incurred by their populations when accessing these services. The modelling approach combined two existing models. For assessing the health benefits the Lives Saved Tool (LiST) was used [13]. LiST is a mathematical modeling tool which allows users to estimate the impact of coverage change on mortality in low and middle income countries. For assessing risk, an expanded version of the risk component of a model developed by the London School of Hygiene and Tropical Medicine (LSTMH) [14] was created. This model assessed the benefits of routine childhood immunization against the excess risk of

SARS-CoV-2 infections during the COVID-19 pandemic in Africa. This combined modeling approach was applied, and modified in six countries (Afghanistan, Iraq, Morocco, Pakistan, Somalia, and Sudan) from November 2020 to March 2021. This article describes the combined model, the field testing, adaptation and refinement processes as well as some findings.

## Materials and methods

### Overview

The framework for this analysis was based on modelling conducted by Roberton T. et al. [15] (Fig 1).

Based on this framework it is assumed that on the supply side, health care providers, equipment and facilities are diverted to deal with the surge in COVID-19 cases leading to chronic staff shortages at health facilities. With many countries depending on international supplies of pharmaceuticals and medical supplies, interruptions in global transport (closed borders, reduced flights, etc.) can lead to a shortage of RMNCH drugs and supplies. On the demand side, local and national shutdowns could negatively affect economies leading to loss of mobility and reductions in family incomes, and increased prices leading to increased financial and other barriers to access. Furthermore, fear of contracting the virus can hold people back from seeking medical care, especially for maternal and child health care services that are sometimes perceived as non-urgent.

Carrying out a Benefit-Risk analysis of maintaining essential health service coverage during a pandemic consists of several steps. Step one requires an assessment of the disruption caused by the pandemic to different health services. Step two involves the identification of mitigation strategies that may be deployed to restore coverage of essential RMNCH services and an assessment of their effectiveness in counteracting the coverage disruptions caused by the pandemic. Step three comprises an estimate of the risks patients incur accessing health services at

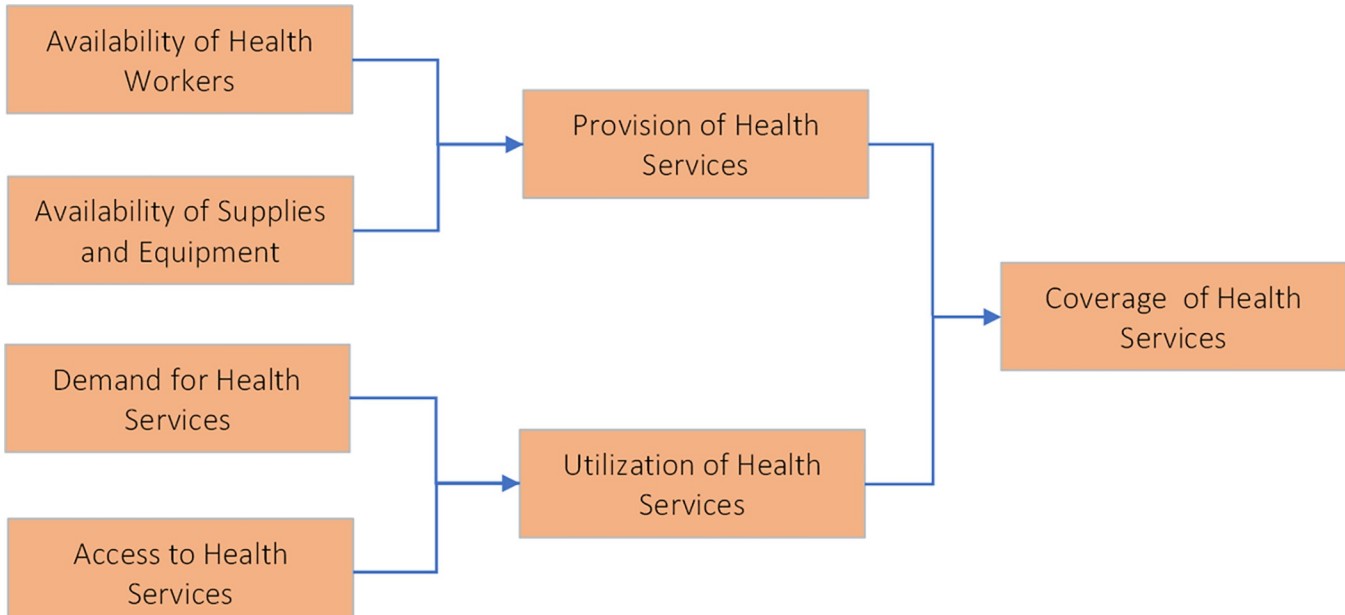

**Fig 1. Framework for the effects of health system components on coverage of health services**\*\*. \*\*Derived from Roberton T. et al. 2020 [15]. Early estimates of the indirect effects of the COVID-19 pandemic on maternal and child mortality in low-income and middle-income countries: A modelling study. Lancet Glob Health 2020; 8: e901–08. (Permission obtained).

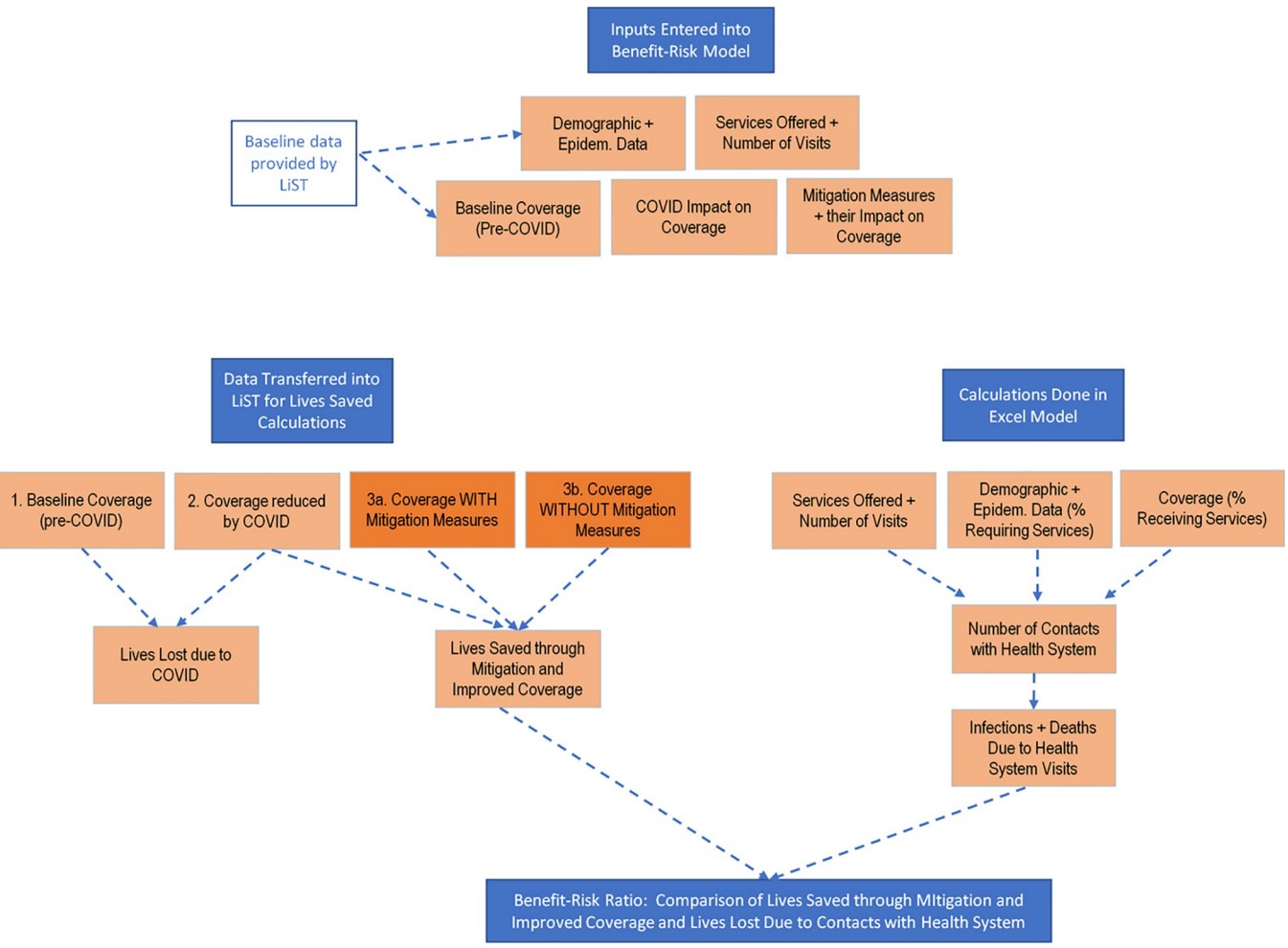

**Fig 2. Model flowchart: Data input and calculations**\*. \*Flowchart describes the overall processes used to input data and enter data into models, and to calculate benefit of maintaining services, risk of infection, and final benefit risk ratio.

facilities, with a particular focus on excess infections caused by contact with the health system. The Excel-based Benefit-Risk analysis developed for this modelling exercise compares lives saved through the continued provision of essential health services with lives lost due to SARS-CoV-2 infections acquired accessing these services, and identifies the most effective interventions (i.e., those that save the most lives.) assuming mitigation strategies succeed to maintain coverage of those interventions.

The modelling approach was based on two pre-existing models. An extensive review of these two models was recently documented including the details of model structure, calculations used, strengths and weaknesses of each and how they were used to assess the indirect impact of COVID-19 pandemic on RMNCH [9]. The specific procedures used to enter the data in these two models and conduct the analysis are shown in the flowchart (Fig 2).

## Inputs entered into model

The following baseline data were collected and entered into the model (Fig 2):

• Baseline demographic and epidemiological data (including household composition and population age structure)

- Baseline coverage of key RMNCH services

- Coverage disruption data for the months that saw a reduction in coverage due to the pandemic. In all countries the largest service reductions were from March to August 2020.

- List of mitigation strategies

- Estimates for the impact of the mitigation strategies (both in terms of their impact on coverage of health interventions as well as their impact on the SARS-CoV-2 transmission risk during the provision of these health services)

The baseline data (baseline population, epidemiological and coverage data) was downloaded from the Spectrum platform, a suite of easy-to-use policy models which consists of several software models including LiST [16]. This platform is regularly updated with the most recent population-based survey data from sources such as the UN Population Division, Demographic and Health Surveys (DHS) and UNICEF Multiple Indicator Surveys (MICS) surveys, as well as other UN databases.

The full range of reproductive, maternal, newborn and child health services a country might provide was included in the analysis. In total, the tool analyzed 69 health interventions in seven packages (Table 1).

Coverage disruption caused by the COVID-19 pandemic was estimated through an analysis of countries' monthly health service statistics//Health Management Information System (HMIS) data. Due to the annual nature of the LiST model it was not possible to enter disruption rates by month, thus for application in countries (see below) an average disruption rate based on the 6 months (March to August 2020) during which services were disrupted was applied to the entire year. This was done for two reasons. One was to exclude the months prior to the pandemic. Two was to exclude the latter months of the year in which mitigation measures may have been in place, so that we estimate what 2020 might have looked like had no mitigation measures been implemented.

Mitigation measures to maintain essential health services were initially taken from the WHO interim guidelines on maintaining essential health services during the pandemic [12].

The strategies were sorted into several categories (S1 Table):

1. Hygiene and Distancing Measures

2. Measures affecting number of visits required to provide care (bundling of visits, provision of several months of supplements or contraceptives during one visit)

3. Measures shifting health care to community providers, tele-health or outreach teams

4. Measures strengthening health worker capacity (increasing availability of providers and training in transmission prevention)

5. Measures to strengthen the supply chain of commodities and medical supplies

6. Measures to promote guidelines and messaging specific to the context of SARS-CoV-2

This list was supplemented by strategies developed by countries that participated in the field-testing phase. Each mitigation strategy was then judged on the impact it might have on restoring coverage of the intervention it was targeting (e.g., antenatal care, immunization) based on consultation with country experts and the available literature at that time [17–21].

## Calculation of Lives Lost and Saved in LiST

Baseline coverage and reduced coverage levels were entered into the LiST model to calculate lives lost due to pandemic-induced coverage reductions for 2020. As all countries saw service

**Table 1. Health service packages included in the analysis, interventions linked to each package and example of proxy indicators used.**

| Healtd Service Package | Specific Interventions | Example of Proxy Indicators |
|---|---|---|
| Family Planning | Pills | Contraceptive Pills Coverage |
| | Condoms | |
| | Injectables | |
| | Implants | |
| | Intra Uterine Device—IUD | |
| | Female Sterilization | |
| | Male Sterilization | |
| | Traditional Methods | |
| Antenatal Care | TT—Tetanus toxoid vaccination | Proportion of pregnant women with 4 or more Antenatal Care Visits |
| | IPTp—Intermittent preventive treatment of malaria during pregnancy | |
| | Syphilis detection and treatment | |
| | Calcium supplementation | |
| | Iron and folate supplementation in pregnancy | |
| | Multiple micronutrient supplementation in pregnancy | |
| | Balanced energy supplementation | |
| | Hypertensive disorder case management | |
| | Diabetes case management | |
| | Malaria case management | |
| | MgSO4 management of pre-eclampsia | |
| Delivery Care&EmOC | Health Facility Delivery | Coverage of Health Facility Delivery |
| | Clean birth environment | |
| | Manual removal of placenta | |
| | MgSO4 management of eclampsia | |
| | Antibiotics for preterm or prolonged PROM | |
| | Parenteral administration of antibiotics | |
| | Assisted vaginal delivery | |
| | Active management of third stage of labor (AMTSL) | |
| | Removal of retained products of conception | |
| | Induction of labor for pregnancies lasting 41+ weeks | |
| | Antenatal corticosteroids for preterm labor | |
| | Maternal sepsis case management | |
| | Safe abortion services | |
| | Post abortion case management | |
| | Cesarean delivery | |
| | Blood transfusion | |
| | Ectopic pregnancy case management | |
| Newborn Care | Immediate drying and additional stimulation | Neonatal resuscitation rates |
| | Thermal protection | |
| | Clean cord care | |
| | Neonatal resuscitation | |
| | Case management of premature babies | |
| | Kangaroo-Mother Care (KMC) | |
| | Full supportive care of prematurity | |
| | Case management of neonatal sepsis/pneumonia | |
| | Oral antibiotics for neonatal sepsis | |
| | Injectable antibiotics for neonatal sepsis | |
| | Full supportive care for neonatal sepsis/pneumonia | |

*(Continued)*

**Table 1.** (Continued)

| Health Service Package | Specific Interventions | Example of Proxy Indicators |
|---|---|---|
| Breastfeeding | Early Initiation of Breastfeeding | Proportion of early Initiation of Breastfeeding |
| | Complementary feeding—Education only | |
| | Complementary feeding—Supplementary feeding + Education | |
| Vaccines | BCG—Single dose | Polio three doses coverage |
| | Polio—Three doses | |
| | Pentavalent | |
| | Diphtheria Pertussis Tetanus—Three doses | |
| | Haemophilus Influenzae (Hib)—Three doses | |
| | Hepatitis B—Three doses | |
| | Pneumococcal—Three doses | |
| | Rotavirus—Two doses | |
| | Meningococcal A—Single dose | |
| | Malaria vaccine—Three doses | |
| | Measles—Single dose | |
| Child Health | Vitamin A supplementation | Oral rehydration solution for diarrhea or Antibiotics for pneumonia |
| | Zinc supplementation | |
| | ORS—Oral Rehydration Solution | |
| | Antibiotics for treatment of dysentery | |
| | Zinc for treatment of diarrhea | |
| | Oral antibiotics for pneumonia | |
| | Vitamin A for treatment of measles | |
| | ACTs—Artemisinin compounds for treatment of malaria | |

disruptions for only half of the year, and LiST only calculates lives lost for the whole year, lives lost results generated from LiST for this model were divided in half to calculate lives lost for the six months of service disruption. Then, using the reduced coverage as baseline, two scenarios were run—the first one assuming that no mitigation measures would be pursued and coverage would remain low and the second one assuming that mitigation measures would be implemented, which in turn would lead to a restoration of coverage levels. The difference in the number of lives saved at the different coverage levels was assumed to be attributable to the mitigation measures.

## Calculations of risk

The assessment of risk of SARS-CoV-2 infection and mortality incurred by health care providers and patients accessing the care (due to the employment of the mitigation strategy) was estimated by adapting and expanding the approach employed by the LSTMH [14] in their study on the risks and benefits of routine childhood immunization during the COVID-19 pandemic. Their approach estimated the excess risk incurred by women as well as children and their caretakers in their quest to receive immunization services at a health facility during the COVID-19 pandemic (this includes the risk of infection during their travel to the facility, waiting and receiving services from a health care provider). The methodology was expanded to cover family planning, maternal, newborn and child health care by estimating the number of health care contacts required per intervention. Services were divided into two categories–ambulatory care and services requiring hospitalization–each carrying different risks of potential infection. They were also assessed on the number of persons exposed to that risk (e.g. antenatal care visits only

exposed the pregnant woman, while child health care visits put both child and caretaker at risk).

Finally, the lives lost and saved from LiST were copied into the Excel model and the excess risk of patients getting infected with SARS-CoV-2, transmitting the disease to their family and the potential for death because of SARS-CoV-2 infection was calculated (S2 Table, and S1 and S2 Text)).

## Country application

The model was field tested, modified, adapted and applied in six countries (Afghanistan, Iraq, Morocco, Pakistan, Somalia, and Sudan) from November 2020 through March 2021. Country teams participated in weekly workshops, in which the methodology was explained and guidance as to what data needed to be collected was imparted. The tool comes with instructions and a set of data collection sheets used for collating all the data necessary to run the model (S2 Table and S1 Text)). After a first workshop that provided an overview of the entire process, each subsequent workshop concentrated on a specific step in the modelling approach. The steps included an assessment of the disruption caused to different health services by the pandemic, the identification of mitigation strategies and an assessment of their effectiveness in counteracting the coverage disruptions, and finally the estimation of the risks health care providers and patients incur accessing health services at facilities.

For the assessment of disruption, countries were asked to gather monthly health service utilization data for 2019 and 2020 for RMNCH services. This was based on WHO guidance for using routine health information for analyzing health service disruption [22]. As data collection was completed in early 2021 and updated as late as March 2021, all countries were able to gather data from Jan 2019 through December 2020 thus potential issues related to delayed or incomplete reporting were lessoned. Recognizing that some HMIS systems may have data quality issues, it was assumed that the data quality issues would be consistent across the years, and it was thus considered possible to calculate relative reductions in service delivery. Most countries saw coverages declines due to the pandemic starting in April of 2020, with disruption continuing over the next three to six months, after which most countries observed some recovery (probably due to a mix of a slowing down of the spread of SARS-CoV-2 as a result of general mitigation measures such as lockdowns and mask mandates as well as specific mitigation measures taken to restore health service coverage). To estimate the average disruption caused by COVID, we only included in the analysis those months (March to August 2020) in which coverage levels were below pre-pandemic levels (the affected months in 2020 were compared to the same months in the year 2019 to account for seasonal variations in disease patterns and health service use). Key indicators (e.g. % of women receiving four or more antenatal care visits, or % of women delivering with a skilled birth attendant) were used to estimate service disruption and then applied to the respective package (i.e., family planning, antenatal care, delivery care, etc) (Table 1). If a country had HMIS data for more than one indicator in a package (i.e., the country had data for institutional delivery and C-section delivery), disruption was calculated based on the average disruption seen across those indicators.

All countries experienced disruption to RMNCH services during the COVID-19 pandemic. Based on the estimates using monthly health service statistics and the processes described above, RMNCH service disruptions varied between countries (Table 2).

For identification of mitigation strategies and their impact on health service coverage rates, as previously described, each country team collated information on what mitigation activities were taking place and each mitigation strategy was judged on the impact it might have on restoring coverage based on the limited scientific evidence available on what might work in

**Table 2. Estimated average coverage disruptions from March to August 2020 applied to the year of 2020 by service package by country and included in the analysis.**

| Service Package | COVID-19 impact on coverage, by country (% decrease in coverage) | | | | | |
|---|---|---|---|---|---|---|
| | Afghanistan | Morocco | Iraq | Pakistan | Somalia | Sudan |
| Family Planning | 23% | 36% | 21% | 30% | 15% | 2% |
| Antenatal Care | 24% | 21% | 38% | 31% | 22% | 13% |
| Delivery Care | 14% | 14% | 4% | 19% | 36% | 9% |
| Newborn Care | 14% | 16% | 19% | 18% | 36% | 19% |
| Breastfeeding | 4% | – | 3% | 18% | – | – |
| Vaccines | 18% | 17% | 23% | 26% | 6% | 5% |
| Child Health | 9% | 17% | 0% | 33% | 18% | 9% |

health settings [17–21], as well as expert opinion (S1 Table). Some mitigation measures were estimated to improve coverage due to the possibility that the mitigation measure would increase patients' confidence that a visit to a health facility would be safe, such as visible hygiene and social distancing measures at health facilities or education campaigns reassuring the population that it was safe to visit health facilities and stressing the importance of continued care seeking, even during the pandemic. Other mitigation measures were thought to increase health care coverage by increasing availability of health work force (by involving community-level health workers or task shifting) and ensuring a reliable supply chain. Due to the challenges of calculating the impact of mitigation measures, a concerted effort was made to keep the impact estimates for individual mitigation measures conservative and the model included control features to ensure that overall coverage assumptions stayed within a defined range (e.g., coverage after mitigation measures were implemented was not allowed to be more than 2% above pre-pandemic coverage levels).

Information provided by country teams to calculate risk of infection included the current data on the COVID-19 epidemic in their respective country, estimated proportion of the population requiring each service and the information that had already been provided on coverage and number of visits required to access specific services (S2 Text).

Each country team were provided a template to write a report and each team wrote a final report with all findings.

## Results

The overall Benefit-Risk Ratio in the six countries analyzed was found to be between 13.7 and 79.2, which means that for every 13.7 to 79.2 lives gained due to increased RMNCH service coverage, there was one loss of a life related to COVID-19 (Fig 3). More specifically, for all countries and for all health packages, the benefit risk ratio was above 1, which means that maintaining services saved more lives than were lost due to additional deaths caused by SARS-CoV-2 infections acquired during contacts with the health system.

The number of lives saved estimated depended on the size of the population, the number of services included in the analysis, the extent of disruption of essential RMNCH services caused by COVID-19, and the number and effectiveness of mitigation measures implemented. Countries that had larger service disruptions and implemented more mitigation measures demonstrated more lives saved.

In terms of benefit-risk ratios by health service package, newborn and delivery care had the highest ratios, i.e., many more lives were saved through restored coverage than were lost due to excess SARS-CoV-2 infections and deaths due to contact with the health system for those services. The benefit-risk ratio for antenatal care was lower as many lives saved through the

**Risk-Benefit Results, Afghanistan**

| Service Package | Lives saved through mitigation measures | Lives lost through added COVID-19 Infections | Benefit-Risk Ratio |
|---|---|---|---|
| Family Planning | 14 | (1) | 15.7 |
| Antenatal Care | 274 | (8) | 34.3 |
| Delivery Care | 1,208 | (14) | 88.5 |
| Newborn Care | 1,995 | (10) | 194.7 |
| Breastfeeding | 1,603 | (1) | 2,978.0 |
| Vaccines | 600 | (101) | 5.9 |
| Child Health | 2,853 | (174) | 16.4 |
| Total | 8,547 | (309) | 27.7 |

**Risk-Benefit Results, Pakistan**

| Service Package | Lives saved through mitigation measures | Lives lost through added COVID-19 Infections | Benefit-Risk Ratio |
|---|---|---|---|
| Family Planning | 191 | (5) | 38.4 |
| Antenatal Care | 3,357 | (107) | 31.3 |
| Delivery Care | 4,462 | (85) | 52.4 |
| Newborn Care | 23,198 | (50) | 466.6 |
| Breastfeeding | 35,752 | (25) | 1,409.5 |
| Vaccines | 5,388 | (476) | 11.3 |
| Child Health | 20,118 | (520) | 38.7 |
| Total | 92,466 | (1,269) | 72.9 |

**Risk-Benefit Results, Iraq**

| Service Package | Lives saved through mitigation measures | Lives lost through added COVID-19 Infections | Benefit-Risk Ratio |
|---|---|---|---|
| Family Planning | 7 | (6) | 1.1 |
| Antenatal Care | 92 | (15) | 6.2 |
| Delivery Care | 192 | (4) | 50.3 |
| Newborn Care | 2,306 | (3) | 783.9 |
| Breastfeeding | 425 | (5) | 82.5 |
| Vaccines | 214 | (72) | 3.0 |
| Child Health | 34 | (134) | -- |
| Total | 3,270 | (239) | 13.7 |

**Risk-Benefit Results, Morocco**

| Service Package | Lives saved through mitigation measures | Lives lost through added COVID-19 Infections | Benefit-Risk Ratio |
|---|---|---|---|
| Family Planning | 8 | (5) | 1.7 |
| Antenatal Care | 64 | (4) | 17.5 |
| Delivery Care | 261 | (11) | 22.9 |
| Newborn Care | 1,125 | (10) | 116.1 |
| Breastfeeding | -- | -- | -- |
| Vaccines | 212 | (74) | 2.8 |
| Child Health | 50 | (6) | 8.7 |
| Total | 1,720 | (108) | 16.0 |

**Risk-Benefit Results, Somalia**

| Service Package | Lives saved through mitigation measures | Lives lost through added COVID-19 Infections | Benefit-Risk Ratio |
|---|---|---|---|
| Family Planning | 16 | (1) | 25.7 |
| Antenatal Care | 484 | (5) | 104.8 |
| Delivery Care | 71 | (6) | 11.5 |
| Newborn Care | 93 | (5) | 17.9 |
| Breastfeeding | -- | -- | -- |
| Vaccines | 135 | (65) | 2.1 |
| Child Health | 965 | (29) | 33.7 |
| Total | 1,764 | (111) | 15.9 |

**Risk-Benefit Results, Sudan**

| Service Package | Lives saved through mitigation measures | Lives lost through added COVID-19 Infections | Benefit-Risk Ratio |
|---|---|---|---|
| Family Planning | 4 | (0) | 8.2 |
| Antenatal Care | 217 | (24) | 8.9 |
| Delivery Care | 369 | (5) | 67.4 |
| Newborn Care | 703 | (3) | 210.3 |
| Breastfeeding | -- | -- | -- |
| Vaccines | 311 | (68) | 4.6 |
| Child Health | 801 | (83) | 9.6 |
| Total | 2,405 | (101) | 23.8 |

**Fig 3. Estimated lives saved as a result of mitigation measures and restored coverage of RMNCH services and benefit-risk ratios, by country for 2020.**

prevention and early detection of complications were not captured under that package, but under delivery and emergency obstetric and newborn care. Vaccinations also had lower benefit-risk ratios, mainly due to the fact that the model assumed that vaccination campaigns would only be disrupted for one year, during which prior vaccinations and herd immunity would continue to provide protection. However, if immunization campaigns are delayed for longer than one-year, deaths would increase significantly in the future. Finally, family planning for all countries, demonstrated modest benefit-risk ratios as disruptions in family planning coverage primarily impacted the number of pregnancies and births in the following year and any deaths due to pregnancy and delivery complications were captured by other health packages, such as ANC and delivery care.

There were many high-impact interventions where the restoration of coverage saved large numbers of lives. In all countries, tetanus toxoid vaccination of pregnant women saved the majority of lives attributed to antenatal care. The restoration of delivery care (which included the early detection and management of pregnancy complications), also saved many maternal and newborn lives. The most effective interventions in the prevention of maternal and neonatal mortality were the provision of C-section and assisted vaginal delivery for complicated births. Also important for saving maternal lives is the active management of the third stage of labor (AMTSL) with oxytocin. A large number of newborns were saved through the restoration of simple interventions such as immediate drying, thermal protection, clean cord care, and neonatal resuscitation. Finally, for child health, restoring services that provide treatment for diarrhea with ORS and pneumonia with antibiotics was lifesaving for many children under 5.

## Discussion

The Benefit-Risk model presented here was a first attempt to understand how mitigation measures, by increasing and potentially restoring pre-pandemic coverage levels of essential RMNCH services and decreasing the risk of COVID transmission at facilities, can save lives. Over time, as more data become available, it will be possible to improve the accuracy of the model.

Overall, with technical support provided by the modeling experts, the exercise proved to be manageable for all six country teams. While the final data entered into the tool was not perfect results across the six countries fell within expected ranges and it was possible to explain differences in benefit-risk ratios among packages and countries by variations in the underlying data (the most significant factors being baseline coverage, size of disruption and recovery in health service utilization, the range of health services included, household and age composition–the latter two affecting chance of transmission and fatality rate). The project provided the opportunity to combine the expertise and experience from several countries, which was particularly useful in an exercise like this that had never been attempted before. Especially countries that joined the exercise later benefitted from the experiences gained in the first rounds, as it was possible to compile a comprehensive list of mitigation measures used across countries with impact estimates as well as a report template.

The results were available through country reports and were presented to ministries of health and other in country health partners. However, since this model was being created and tested while urgent decisions about essential health services had to be made it was not possible to use this information in real time, but future applications of a refined tool for not only the COVID-19 pandemic but other pandemics and emergencies should make such decision-making possible.

Several important limitations must be understood when interpreting results from this Benefit-Risk modelling exercise. To calculate coverage disruption, countries used routinely available health facility data sources, which may contain errors and suffer from data entry delays. They usually also do not cover services offered by private providers and are thus not representative of service coverage for the entire population. However, since our modelling exercise others have examined COVID-19 pandemic essential health services disruption trends using routine health information and have shown disruption trends similar to what we found here [23]. Due to the recent and constantly shifting nature of the COVID-19 pandemic, there was not much empirical evidence available to assist in making the estimates used in this benefit-risk analysis. At the time of the study, no evaluation had been done yet of how mitigation measures affected coverage of essential RMNCH services during the COVID-19 pandemic. Estimates used in this analysis were thus mainly based on expert opinion. Slightly easier, but still not well documented, was the estimation of how mitigation measures might affect the risk of

infection of virus transmission at a health facility. ⁻Since this modelling exercise was under-taken some evidence, for example regarding immunization, that service disruption may have recovered more quickly than anticipated [24]. However, this should not significantly impact our findings as we assumed only one year of disruption and that herd immunity would provide protection. This benefit-risk analysis used the LiST model in a new way, as LiST was originally designed to model the impact of gradual coverage scale-ups, not short-term disruptions [13]. The annual nature of the LiST model also imposed some limitations on the analysis, as it was not possible to enter disruption rates by month, an average disruption rate had to be used and applied to the entire year. The exercise was technically relatively advanced, requiring countries to navigate an Excel model as well as enter and extract information from Spectrum/LiST, a tool that might not be familiar to the average health expert.

Not considered in the interpretation of the findings, was the possibility that there might have been a decrease in the need for health services during the pandemic as well. This applies specifically to child health indicators, especially antibiotics for ARI–one might reasonably hypothesize that the same mitigation measures put into place to limit the spread of SARS-CoV-2 would also decrease transmission of respiratory infections. The same might also be true for diarrhea via school closures, social distancing, etc., though perhaps to a smaller degree. Assessing this complex interplay between demand and service position was beyond the scope of this study.

Overall, this benefit-risk analysis underscored the importance and value of maintaining coverage of essential health services even during public health emergencies, including the recent COVID-19 pandemic. In all cases and for all services, the benefit of maintaining essential health services far exceeded the risks associated with additional SARS-CoV-2 infections and deaths. This benefit-risk analysis illustrated how essential health services can continue to operate during a pandemic and how mitigation measures can reduce SARS-CoV-2 infections and increase coverage of essential health services.

Some modifications will be necessary for this tool to be useful for advocacy, policy, and pro-gram planning in the future. For one, the risk component of the tool will need to be updated to better capture the complex and everchanging nature of the pandemic (second and third waves, new strains, increase in vaccine availability). Also, as more evidence becomes available, it should be possible to confirm and improve the impact estimates for the mitigation measures. A modification of the LiST impact model to allow for more short-term projections would allow for more timely analysis. A refinement of the excel tools to automate the data transfer from and to Spectrum and LiST are also needed.

The update of the risk component and an expansion of the tool to cover more essential health services (such as HIV/AIDS, NCDs) are currently being explored by WHO as the appli-cation of the tool in the areas of RMNCH identified a possible broader use of this tool for all essential health services with possible consideration of inclusion into the Spectrum package. In addition to the use for advocacy to ensure that health managers and policy makers understand the relative importance of maintaining these services to save lives in comparison to preventing transmission of SARS-CoV-2 virus, this modelling approach can potentially be used to help in the allocation of staff and resources to save the largest number of lives, even beyond the emer-gency response for the COVID-19 pandemic but also for other pandemics and emergencies in the future.

## Supporting information

**S1 Table. Mitigation measures and estimated impacts on coverage of essential health inter-ventions as well as COVID transmission risk: Examples of mitigation measures deployed**

by countries with estimated effectiveness data.
(DOCX)

**S2 Table. Data entry form: Benefit-risk model for maintaining essential RMNCAH services in the COVID pandemic.**
(XLSM)

**S1 Text. Benefit-risk analysis of maintaining coverage of essential health interventions during the COVID pandemic: User guidelines.**
(DOCX)

**S2 Text. Key formulas of risk-benefit model.**
(DOCX)

## Acknowledgments

The authors alone are responsible for the views expressed in this article and they do not necessarily represent the views, decisions or policies of the institutions with which they are affiliated.

This work would not have been possible without the support of the following: Dr Maha El Adawy–Director, Department of Healthier Populations, Dr Arash Rashidian–Director, Science, Information and Dissemination, Late Dr Ramez Mahaini–Coordinator Maternal and Child Health, Dr Karima Gholbzouri–Regional Advisor, Reproductive and Maternal Health, Dr Jamela Al Raiby- Regional Advisor, Child and Adolescent Health, Dr Henry Doctor–Coordinator- Science, Information and Dissemination, Mrs. Mae Elzaby–Programme Assistant, Reproductive and Maternal Health Unit (all WHO Eastern Mediterranean Regional Office);the Ministry of Public Health, Afghanistan; Dr Raghad Abdul Redha, Dr Maha Rasheed, Dr Majeda Ahmed, Dr Lujain Muhammed, Dr Tayser Salah, Dr Majd Basim Mohamed (Reproductive health and school health department, Ministry of Health, Iraq); Dr Arjwan Marwan Shaban (Vital and health statistic department/Ministry of Health, Iraq); Dr Abdelhakim Yahiane, Dr Sanaa Elomrani, Dr Mohammed Benazouz, Dr Hafida Yarthaoui, Dr Aziza Yaghfouri, Anouar Talouizte (Population Directorate—Ministry of Health–Morocco); Dr Malik Muhammad Safi, Dr Sarah Ashraf (Ministry of National Health Services, Regulation & Coordination, Pakistan); Dr Palitha Mahipala, Ms Ellen Thom (WHO country office- PakistanMr Usman Bashir (WHO Consultant, Pakistan); Dr. Naima Abdulkadir—Federal Ministry of Health, Somalia, Dr. Abdulmunim Mohamed–WHO Country Office, Somalia; Dr. Dalya Eltayeb, Khalid Mahagoub Abdalaziz Ahmed, Osama Mohamed Ismail, Mawahib Elfadil Saeed, Bahga Siddeq, Muhammed Abdelghani Omer (Federal Ministry of Health Sudan); Dr Elamin Maison, Dr Hiba Hussein (WHO Country office Sudan); Dr. Abdelmalik M. Hashim (WHO Consultant Sudan), Mohamed Khalid Abbas–Math Modelling/ IEND, University of Khartoum, Dr. Farah Ibrahim (UNICEF Sudan) and Dr. Rania Hassan (UNFPA Sudan).

We thank Kate Gilroy (JSI), William Weiss (USAID), Tim Roberton (Johns Hopkins University), Jim Ricca (JHPIEGO) and Howard Friedman (UNFPA) for reviewing early versions of the model and providing detailed suggestions for improvements of formulas and assumptions. Kaja Abbas from the London School of Hygiene and Tropical Medicine reviewed the model's expanded risk modeling assumptions. Neff Walker and Victoria Chow (Johns Hopkins University) and Bill Winfrey provided invaluable support interpreting LiST outputs and ensuring that the new model was properly tied to the impact calculations of the LiST model.

## Author Contributions

**Conceptualization:** Eva Weissman, Denise Buchner, Nilmini Hemachandra, Theresa Diaz.

**Data curation:** Nilmini Hemachandra, Khalid Siddeeg, Mohammad Samim Soroush, Ahmed Javed Rahmanzai, Paata Chikvaidze, Zaid Muayad Yassen, Hanan Hasan, Mohamed Berraho, Nouzha Dghoughi, Hachri Hafid, Raza Mahmood Zaidi, Wahaj Zulfiqar, Sayema Awes, Atiya Aabroo, Qudsia Uzma, Mohamed Abubakar Hagi Mohamed Fiidow, Abdulkadir Wehliye Afrah, Abdullahi Abdulle Ali, Abdulazim Ali Awadalla, Muntasir Mohammed Osman EL Hassan, Esmehan Babeker Elkheir.

**Formal analysis:** Eva Weissman, Denise Buchner, Mohammad Samim Soroush, Ahmed Javed Rahmanzai, Paata Chikvaidze, Zaid Muayad Yassen, Hanan Hasan, Mohamed Berraho, Nouzha Dghoughi, Hachri Hafid, Raza Mahmood Zaidi, Wahaj Zulfiqar, Sayema Awes, Atiya Aabroo, Qudsia Uzma, Mohamed Abubakar Hagi Mohamed Fiidow, Abdulkadir Wehliye Afrah, Abdullahi Abdulle Ali, Abdulazim Ali Awadalla, Muntasir Mohammed Osman EL Hassan, Esmehan Babeker Elkheir.

**Funding acquisition:** Theresa Diaz.

**Methodology:** Eva Weissman, Denise Buchner, Theresa Diaz.

**Project administration:** Nilmini Hemachandra.

**Supervision:** Eva Weissman, Nilmini Hemachandra, Khalid Siddeeg, Theresa Diaz.

**Validation:** Eva Weissman, Denise Buchner, Khalid Siddeeg, Mohammad Samim Soroush, Ahmed Javed Rahmanzai, Paata Chikvaidze, Zaid Muayad Yassen, Hanan Hasan, Mohamed Berraho, Nouzha Dghoughi, Hachri Hafid, Raza Mahmood Zaidi, Wahaj Zulfiqar, Sayema Awes, Atiya Aabroo, Qudsia Uzma, Mohamed Abubakar Hagi Mohamed Fiidow, Abdulkadir Wehliye Afrah, Abdullahi Abdulle Ali, Abdulazim Ali Awadalla, Muntasir Mohammed Osman EL Hassan, Esmehan Babeker Elkheir.

**Writing – original draft:** Eva Weissman, Denise Buchner.

**Writing – review & editing:** Eva Weissman, Denise Buchner, Nilmini Hemachandra, Khalid Siddeeg, Mohammad Samim Soroush, Ahmed Javed Rahmanzai, Paata Chikvaidze, Zaid Muayad Yassen, Hanan Hasan, Mohamed Berraho, Nouzha Dghoughi, Hachri Hafid, Wahaj Zulfiqar, Sayema Awes, Atiya Aabroo, Qudsia Uzma, Mohamed Abubakar Hagi Mohamed Fiidow, Abdulkadir Wehliye Afrah, Abdullahi Abdulle Ali, Abdulazim Ali Awadalla, Muntasir Mohammed Osman EL Hassan, Esmehan Babeker Elkheir, Theresa Diaz.

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
