## [Decision Letter · Decision Letter 0]

7 Sep 2021

PGPH-D-21-00183

Risk-Benefit Analysis of Maintaining Essential Reproductive, Maternal, Newborn, and Child Health (RMNCH) Services Against Risk of Covid-19 Infection

Dear Dr. Diaz,

Thank you for submitting your manuscript to PLOS Global Public Health. After careful consideration, we feel that it has merit but does not fully meet PLOS Global Public Health’s publication criteria as it currently stands. Therefore, we invite you to submit a revised version of the manuscript that addresses the points raised during the review process.

We look forward to receiving your revised manuscript.

Kind regards,

Hannah Tappis, DrPH, MPH

Academic Editor

Journal Requirements: 

Reviewers' comments:

Reviewer's Responses to Questions

**Comments to the Author**

1. Does this manuscript meet PLOS Global Public Health’s publication criteria? Is the manuscript technically sound, and do the data support the conclusions? The manuscript must describe methodologically and ethically rigorous research with conclusions that are appropriately drawn based on the data presented.

Reviewer #1: Yes

Reviewer #2: Partly

2. Has the statistical analysis been performed appropriately and rigorously?

Reviewer #1: I don't know

Reviewer #2: I don't know

3. Have the authors made all data underlying the findings in their manuscript fully available (please refer to the Data Availability Statement at the start of the manuscript PDF file)?

Reviewer #1: No

Reviewer #2: No

4. Is the manuscript presented in an intelligible fashion and written in standard English?

Reviewer #1: Yes

Reviewer #2: Yes

5. Review Comments to the Author

Reviewer #1: Overview:

This interesting article describes the experience of planning and conducing a risk-benefit analysis for essential RMNCH services during the COVID-19 pandemic in select countries in 2020. As the authors describe, the article aims to both describe the technical aspects of the modeling framework used as well as reflect on the experience of conducting this exercise during the pandemic. Achieving both of these important aims in a single manuscript is certainly an ambitious undertaking. Below, I have tried to focus my comments on areas where some additional methodological detail or description might help to strengthen the manuscript’s ability to achieve these interrelated goals.

Major comments:

1. To improve the clarity of the manuscript, reorganization of the introduction and methods would be helpful. As currently written, the manuscript mixes together background information (i.e. a general description of mechanisms for essential health service coverage disruption in lines 95-106, or a description of potential mitigation strategies and their impact in lines 116-122) with more detailed methodological description that is specific to the analysis presented in this manuscript. I think that the manuscript would be clearer if some of this information were to be moved to the introduction section, so that the introduction contains the background information that sets the stage for the risk-benefit analysis introduced here, and the methods section clearly describes what was done for this particular analysis.

2. Within the methods, a clearer description of the modeling/analytic framework at an earlier stage in the methods section would be helpful. Currently, this information is somewhat scattered throughout the methods. For instance, there is a brief mention of LiST on lines 132-135, a list of key model inputs and other pieces of the model on lines 154-161, and then a very brief description of an Excel model and LiST interacting in lines 216-221.

I might suggest that providing a high-level overview of the modeling framework early in the methods section would be helpful, perhaps accompanied by a modeling flowchart and/or analytic diagram that shows the key inputs into the model as well as more detail about the modeling steps (i.e. what is done by the Excel tool and what is done by LiST). This would help to orient the reader in preparation for more detailed discussions about the data and assumptions used to inform the model later on.

Last, I think that the description of the Excel model and its interactions with the LiST model (lines 216-221) could benefit from some expanded detail. In particular, it would be useful to be specific about the calculations used here for the coverage levels under different intervention scenarios, and what the lives lost and saved from LiST represent (i.e. what time period was evaluated – just 2020, or 2020-2021, or beyond). From the relatively brief description given in the methods section, it is difficult to understand exactly how all of the various data inputs and assumptions were synthesized to produce the final results presented here.

3. It would be helpful to more clearly describe how several of the input data estimates were derived. My sense from reading the article is that the country teams made the ultimate decisions about which input data to use, though guided by expert opinion / recommendations, specific quantitative analyses performed for this exercise (i.e. of service disruption magnitude), or other sources (i.e. Spectrum). In some sections of the methods, however, this process is not entirely clear.

a. For the baseline data section (lines 163-164), this is clearly laid out – each country reviewed data from Spectrum but had the option to use updated or better-quality data at their discretion.

b. For the disruption data section, there is a description of a quantitative analysis that was used to estimate disruptions (lines 171-183). It seems, however, that there was also some ability for countries to make individual decisions about the best way to assess service disruptions in their country (i.e. lines 183-184). It would be helpful to better understand how country expertise and this quantitative analysis were combined to reach final estimates of coverage reduction. (I have several technical questions about the quantitative analysis used to inform disruption estimates in a separate comment below).

c. For the mitigation strategies (lines 192-213), there is a statement that “potential impacts of these mitigation measures were evaluated with a view to [three potential impacts on health outcomes]”. In Annex 2, I see suggested and country-specific impact estimates for the various mitigation measures. It would be helpful for the authors to explain how this process worked – i.e. who came up with the “suggested” estimates and how, and how were these suggested estimates related to the final country-specific impact estimates? Based on the discussion section, my guess is that there was some sort of expert consensus process used for the suggested estimates, and then country teams could choose to adopt or modify those based on their own assumptions. Still, it would be helpful to more explicitly describe this process in the methods section.

4. Lines 177-180: The approach that the authors take to estimate disruptions due to COVID for essential health services generally seems reasonable, although I have a few specific questions about the approach used here.

a. First, were there any months in 2019 for which there were unusual fluctuations in service provision in the countries in question (i.e. due to shortages or stockouts of equipment or supplies, health worker strikes, etc.?). If so, were those months additionally excluded from the analysis?

b. Second, did the authors account for expected changes in any indicators between 2019 and 2020 due to, e.g., changes in population size? If a country experiencing population growth, for instance, simply providing the same number of ANC visits, etc. may represent a decrease in the proportion of women receiving care. If any adjustments were made, it would probably be worth mentioning those here; if not, perhaps the authors could mention if/why this effect was thought to be negligible.

c. Were the authors able to assess whether any of the observed changes in service delivery during the pandemic may have resulted from either delayed or incomplete reporting in 2020 compared to 2019? Especially during the early months of the pandemic, one might expect that the same factors that disrupted health services more generally may have made reliable and timely reporting more difficult. From some analyses of vaccine coverage data collected throughout the pandemic, it seems that monthly data reported by many countries was subsequently revised (usually upwards) in later reports, as might be expected with ongoing data collection (especially during the pandemic). Perhaps, however, the HMIS data also contains ancillary data quality indicators that suggested that delayed and/or incomplete reporting wasn’t an issue in 2020. In either case, it may be worth a brief note in the manuscript to address this potential question.

d. For some services – i.e. vaccination – large disruptions early in the pandemic could be at least partially mitigated by concerted catch-up services later in the pandemic. Was this possibility taken into account? Or did the strategy of only using months for which disruptions were reported limit the model’s ability to account for catch-up vaccination, for instance?

e. Last, the authors write that they considered only months for which there were observed disruptions. I’m not quite sure how that translates into the disruption estimates that are later presented in Figure 4. Does this mean that a country seeing a 50% reduction in services in March 2020 and 30% reduction in services in April 2020 – but no subsequent disruptions after April – would be assigned a 40% reduction? Or would this be averaged out with “no-disruption” months from May-December to produce an estimate from March – December 2020 (i.e. an 8% reduction in total between March – December 2020)? Some clarification around how to interpret these indicators would be helpful. For instance in Figure 4, there is a table that presents a comparison of the level of service disruption for various indicators. What does what “COVID-19 impact on coverage, by country” means in this table? Is this the impact on coverage for the full year of 2020 (i.e. including January and February 2020)? For just the pandemic months (i.e. March – December)? Or for just the months in which there were reported service disruptions?

5. In the discussion, the authors reflect on the experience of country teams in using this tool to conduct a risk-benefit analysis. For instance, on lines 288-301, the authors describe that the exercise was manageable and produce what seemed like reasonable response, and that ongoing learning throughout the process was helpful to countries that joined the exercise later. It would be helpful, however, to also know whether or not the exercise was used to guide any policy decisions. If so, what sorts of decisions were informed; if not, what barriers may have limited the perceived utility of these results or their uptake?

Minor comments:

1. I may have missed this, but what was the time frame that countries were exploring during this analysis? I see that the field testing took place from September – December 2020. Were the country teams analyzing what was currently happening (i.e. in 2020), or were they looking forward (i.e. 2021 and/or additional years afterwards?)

2. Lines 89-93: The authors state that the Excel-based risk-benefit analysis “identifies the most effective mitigation interventions and strategies”. I see in the results that the authors discuss which interventions / health service packages were most influential in saving lives in this modeling exercise, but I don’t actually see an analysis of which *mitigation* strategies were most effective (e.g. if temperature screening is more effective than social distancing or vice versa). Is this a misstatement, or have I misinterpreted the analyses presented here?

3. Figure 3: For some of these indicators, it seems possible that the demand for the health service may have declined during the pandemic as well. (I am thinking here specifically of the child health indicators, especially antibiotics for ARI – one might reasonably hypothesize that the same mitigation measures put into place to limit the spread of COVID would also decrease transmission of respiratory infections. The same might also be true for diarrhea via school closures, social distancing, etc., though perhaps to a smaller degree.) It seems that assessing this complex interplay between demand and service position is probably beyond the scope of this study, but I wonder if the authors could comment on whether this is a limitation for the present study and/or if this was taken into account when these disruptions were entered into LiST.

4. Some of the reported disruptions in Figure 4 are quite a bit larger than other data sources suggest. For instance, comparing the recently released WUENIC estimates and data to Figure 4 for vaccination coverage (here using DTP3 from WUENIC from https://immunizationdata.who.int/pages/coverage/dtp.html?CODE=AFG+MAR+IRQ+PAK+SOM+SDN&ANTIGEN=DTPCV3&YEAR=), AFG reported 86.7% coverage in 2019 (WUENIC estimate: 72%) and 85.3% in 2020 (WUENIC estimate: 70%). Both the WUENIC estimates and official country-reported data therefore suggest a much smaller disruption than the 18% disruption indicated here. Do the authors have any insight into the reason for these differences? (I appreciate that this is quite new data and that it’s not clear what the true level of disruption was over the course of 2020).

5. Lines 285-287: “The annual nature of the LiST model also imposed some limitations on the analysis.” Could the authors provide a bit more detail about what these limitations might be and how they should affect the reader’s interpretation of the results?

6. Line 212 refers to Annex 3 – should this be Annex 2? In Annex 2, I’m also not sure how to interpret the rows in which only some countries have their own estimated effect sizes. For instance, “Implement a requirement that patients bring their own mask” has only a suggested effect size but no country-specific effect sizes. Does this mean that no countries implemented this measure? If so, I would suggest adding a note to the table to make this clear (i.e. impact estimates are only given for countries that included this strategy as part of their mitigation efforts).

Reviewer #2: The submitted manuscript, Risk-Benefit Analysis of Maintaining Essential Reproductive, Maternal, Newborn, and Child Health (RMNCH) Services Against Risk of COVID-19 Infection provides insights into one approach countries have taken in assessing the trade-offs between death due to the novel coronavirus infections and essential health services, with the support of the World Health Organization. For this reason, this reviewer finds this manuscript of value for the global health community; particularly, of interest to those potential users of such a tool.

However, the current state of the manuscript leaves the reader with major questions mostly due to poorly explained figures and the lack of supporting evidence supposedly available in an annex, but was not provided in this review. While the authors list having adapted the work of Roberton et al. and Abbas et al. few to no clear assumptions or sensitivity analyses were provided.

Major comments:

- Figure 1 in this manuscript is not sourced from the original authors (Roberton et al.) and no indication is provided if the current authors were given rights to republish.

- In general, figures are not adequately captioned or explained in the manuscript. As an example, Figure 1, Figure 2, nor Figure 4 are not mentioned in the manuscript and lack captions for the readers to understand/interpret their importance.

- Citations are lacking in the entire introduction, though many statements warrant references.

- Annex 2 is references in the manuscript as containing the "detailed explanation of the variable and formulas used" however, Annex 2 seems to be what is referred to as Annex 3 in the manuscript - "an overview over the most common mitigation strategies and estimates of their impact"

- Annex 2 should be included

- Current "Annex 2" should be relabeled as Annex 3

- Annex 1 in the manuscript is mentioned to have 72 interventions, however the available Annex 1 only contains 69. This should be remedied.

- The limitations of the approach taken is not well addressed as the assumptions underlying the analyses are lacking in the main manuscript.

- Were any sensitivity analyses conducted by country stakeholders? It would be interesting to understand how changes in an assumption or the underlying data might lead to variations in outcomes.

- Manuscript could be strengthened by clearly stating how policy makers and researchers could access this tool and leverage it to adjust for the changing dynamics of the coronaviruses (detla, gamma, etc.)

Minor comments:

For clarity, this reviewer recommends that the authors use either "risk-benefit" or "benefit-risk" and not interchange these as readers may be confused. This is found in both the manuscript text and figures.

6. PLOS authors have the option to publish the peer review history of their article (what does this mean?). If published, this will include your full peer review and any attached files.

**Do you want your identity to be public for this peer review?** For information about this choice, including consent withdrawal, please see our Privacy Policy.

Reviewer #1: No

Reviewer #2: No

---

## [Decision Letter · Decision Letter 1]

9 Dec 2021

PGPH-D-21-00183R1

Benefit-Risk Analysis of Maintaining Essential Reproductive, Maternal, Newborn, and Child Health (RMNCH) Services Against Risk of Covid-19 Infection

Dear Dr. Diaz,

Thank you for submitting your manuscript to PLOS Global Public Health. After careful consideration, we feel that it has merit but does not fully meet PLOS Global Public Health’s publication criteria as it currently stands. Therefore, we invite you to submit a revised version of the manuscript that addresses the points raised during the review process.

We look forward to receiving your revised manuscript.

Kind regards,

Hannah Tappis, DrPH, MPH

Academic Editor

Journal Requirements:

Additional Editor Comments (if provided):

The majority of comments from previous reviewers have been well addressed in the revised manuscript. Please consider outstanding queries regarding assumptions made in estimation of health service disruptions, as well as additional concerns raised by a third reviewer (both attached) - either by providing clear explanations and rationale for assumptions, or discussing implications of model limitations.

Reviewers' comments:

Reviewer's Responses to Questions

**Comments to the Author**

1. If the authors have adequately addressed your comments raised in a previous round of review and you feel that this manuscript is now acceptable for publication, you may indicate that here to bypass the “Comments to the Author” section, enter your conflict of interest statement in the “Confidential to Editor” section, and submit your "Accept" recommendation.

Reviewer #1: (No Response)

Reviewer #3: (No Response)

2. Does this manuscript meet PLOS Global Public Health’s publication criteria? Is the manuscript technically sound, and do the data support the conclusions? The manuscript must describe methodologically and ethically rigorous research with conclusions that are appropriately drawn based on the data presented.

Reviewer #1: Yes

Reviewer #3: Partly

3. Has the statistical analysis been performed appropriately and rigorously?

Reviewer #1: Yes

Reviewer #3: I don't know

4. Have the authors made all data underlying the findings in their manuscript fully available (please refer to the Data Availability Statement at the start of the manuscript PDF file)?

Reviewer #1: Yes

Reviewer #3: Yes

5. Is the manuscript presented in an intelligible fashion and written in standard English?

Reviewer #1: Yes

Reviewer #3: Yes

6. Review Comments to the Author

Reviewer #1: See attached

Reviewer #3: See attached

7. PLOS authors have the option to publish the peer review history of their article (what does this mean?). If published, this will include your full peer review and any attached files.

**Do you want your identity to be public for this peer review?** For information about this choice, including consent withdrawal, please see our Privacy Policy.

Reviewer #1: No

Reviewer #3: No

---

## [Editor Report · Decision Letter 2]

9 Jan 2022

Benefit-Risk Analysis of Maintaining Essential Reproductive, Maternal, Newborn, and Child Health (RMNCH) Services Against Risk of Covid-19 Infection

PGPH-D-21-00183R2

Dear Dr. Diaz,

We're pleased to inform you that your manuscript has been judged scientifically suitable for publication and will be formally accepted for publication once it meets all outstanding technical requirements.

Within one week, you'll receive an e-mail detailing the required amendments. When these have been addressed, you'll receive a formal acceptance letter and your manuscript will be scheduled for publication.

An invoice for payment will follow shortly after the formal acceptance. To ensure an efficient process, please log into Editorial Manager at https://www.editorialmanager.com/pgph/ click the 'Update My Information' link at the top of the page, and double check that your user information is up-to-date. If you have any billing related questions, please contact our Author Billing department directly at authorbilling@plos.org.

Kind regards,

Hannah Tappis, DrPH, MPH

Academic Editor
